# Nano-Drug Design Based on the Physiological Properties of Glutathione

**DOI:** 10.3390/molecules26185567

**Published:** 2021-09-13

**Authors:** Wenhua Li, Minghui Li, Jing Qi

**Affiliations:** Daqing Campus, Harbin Medical University, 39 Xinyang Rd., Daqing 163319, China; zhanzaigudi@126.com

**Keywords:** glutathione, physiological property, nano-drugs, review

## Abstract

Glutathione (GSH) is involved in and regulates important physiological functions of the body as an essential antioxidant. GSH plays an important role in anti-oxidation, detoxification, anti-aging, enhancing immunity and anti-tumor activity. Herein, based on the physiological properties of GSH in different diseases, mainly including the strong reducibility of GSH, high GSH content in tumor cells, and the NADPH depletion when GSSH is reduced to GSH, we extensively report the design principles, effect, and potential problems of various nano-drugs in diabetes, cancer, nervous system diseases, fluorescent probes, imaging, and food. These studies make full use of the physiological and pathological value of GSH and develop excellent design methods of nano-drugs related to GSH, which shows important scientific significance and prominent application value for the related diseases research that GSH participates in or responds to.

## 1. Glutathione Structure

Glutathione (GSH) was discovered by Hopkins in 1921 [1], and is a tripeptide compound formed of glutamic acid, cysteine, and glycine through the peptide bond condensation. Its chemical name is γ-L-glutamyl-L-cysteyl-glycine, and the molecular formula is C_10_H_17_O_6_SN_3_ [2]. There are two kinds of glutathione, namely reduced glutathione (GSH) and oxidized glutathione (GSSG). The structure of GSH contains an active reducing group, sulfydryl (-SH), which is easily oxidized and dehydrogenated. Glutathion peroxidase (GSH-Px) can catalyze GSH to GSSG, while glutathione reductase (GSH-R) can use nicotinamide adenine dinucleotide phosphate (NADPH) to catalyze GSSG to GSH. The main active group of GSSG is disulfide bond (-SS-). GSH biosynthesis is directly controlled by the synthase system, rather than like the protein synthesis on the ribosome [3,4,5]. The specific structure and synthesis procedure are shown in Figure 1.

## 2. Physiological Function of GSH

GSH is found in almost every cell of the body [6], and widely presents in a variety of essential organs and tissues, such as blood, liver, and kidney, in which the liver and kidney are the main synthetic, metabolic, and excretory organs of GSH [7]. Generally, GSH plays an important physiological role in organisms, while GSSG needs to be reduced to GSH to achieve physiological activity. GSH maintains the normal function of immune system and has the obvious antioxidant and detoxification effects. Moreover, the unique structure of GSH makes it to be a prominent free-radical scavenger in the body [8,9,10]. Therefore, GSH has the advantages of excellent roles in anti-aging, enhancing immunity, and anti-tumor activity [11,12,13]. When a small amount of H_2_O_2_ is generated in the cell, GSH reduces H_2_O_2_ to H_2_O with GSH-Px, while being oxidized to GSSG. GSSG accepts H^+^ and reduces to GSH with GSH-R, so that the scavenging reaction of free radicals in the body can continue, which protects the structure and function of cell membrane from interference and damage of oxides [14]. In addition, GSH also has a relieving activity on neuronal excitatory intoxication [15], which can be utilized to alleviate the toxic and side reactions caused by chemotherapy in patients with malignant tumors [16].

Although GSH plays an important role in physiological functions, significant limitations remain, including its inability to penetrate cell membranes, easy oxidation, poor stability, and low bioavailability, which dramatically compromise the effectiveness of the treatment in diseases. Nanotechnology is a novel drug delivery technology which makes the biological active substance embedded or modified on the nanomaterials mainly through physical, chemical, and other conjugation methodologies. Nanoparticles formed by encapsulation or self-assembly can not only protect the biological activity of GSH, but also improve its stability and bioavailability. In addition, the strong reduction characteristic of GSH in microenvironment can be used to cleave specific redox responsive nanoparticles in order to achieve effects of controlled-release and target of drugs. Therefore, in this review, we focus separately on the design principles, effects, and potential problems of various nano-drugs based on the physiological property of GSH in different diseases. Moreover, current challenges and future strategies for developing nano-drugs are also discussed from practical application point of view.

## 3. Nano-Drug Delivery Systems

With the high investment and rapid development in recent years, nanotechnology has been applied in all fields of biomedical science and technology [17]. Similarly, nanotechnology provides new approach for drug delivery, especially targeted drug delivery. Targeted drug delivery systems deliver desired drugs to the diseased parts and reduce distribution to normal tissues or cells [18]. Advantages of nanoparticles as drug delivery systems are described as follow: (1) Dissolve insoluble drugs and prevent drug degradation from the body; (2) prolong the circulation time of drugs; (3) exhibit good biocompatibility and biodegradability; (4) possess high drug loading capacity and low toxicity; (4) selectively deliver drugs to therapeutic targets, such as tumor tissue, tumor cells, tumor-associated stromal cells, and suborganelles [19]. So far, numerous materials such as polymers, lipids, and inorganic materials have been developed and used as drug carriers to control the release behavior of drugs [20,21]. In addition, REDOX response stimulation has been highly valued in the treatment of disease and is widely used in nanomedical drug delivery [22,23]. REDOX potentials in microenvironments are multivariable in different tissues and can be used to design REDOX sensitive delivery systems. Therefore, the design and manufacture of glutathione responsive nanoparticles may be a promising approach for targeted drug delivery [24]. 

## 4. Nano-Drug Design for Diabetes Based on the Physiological Properties of GSH

### 4.1. Nano-Drug Design Based on the Role of GSH in Oxidative Stress

Oxidative stress has been confirmed as a predominant pathogenesis for diabetes, and hyperglycemia is a primary risk factor for promoting the production of reactive oxygen species (ROS). There are multiple kinds of ROS, such as superoxide anion (O^2−^), hydrogen peroxide (H_2_O_2_), hydroxyl radical (OH^−^), nitrogen dioxide (NO_2_) and nitric oxide (NO) free radicals, and so on [25]. A series of antioxidant in the normal body, include vitamin A, vitamin C, vitamin E, GSH, superoxide dismutase (SOD), GSH-Px, and GSH-R, etc., [26]. Among them, GSH is an important member of the body’s endogenous antioxidant. It has the advantages of scavenging free radicals, alleviating damage and maintaining redox equilibrium in cells [27]. When the body is attacked by free radicals, GSH can be used as a direct scavenger of free radicals, a co-substrate of GSH-Px, a cofactor of enzymatic reaction, and a conjugate of many endogenous reactions to improve oxidative stress and delay the development of diabetes [28].

Notably, many researchers have designed nano-drugs for treatment of diabetes and complications based on the physiological role of GSH in oxidative stress. Wei Wang et al. [29] designed a novel antioxidant glutathione liposome (GSH-LIP) to apply in therapy of diabetic nephropathy. GSH-LIP could not only improve the bioavailability of GSH, but also remove the excess ROS induced by oxidative stress and improve the antioxidant capacity. Xiao et al. [30] developed a delivery system composed of enteric Eudragit L100-cysteine/reduced glutathione nanoparticles (Eul-cys/GSH NPs) for oral delivery of insulin. They found that Eul-cys/GSH NPs could promote the intestinal absorption of insulin, and prolong the time of blood sugar reduction, which suggested that Eul-cys/GSH NPs might be a promising delivery system for diabetic therapy. The above nano-drug designs of GSH were that drugs were encapsulated in phospholipids or amphiphilic materials, such as liposome, micelles, as shown in Figure 2A. Kuan et al. [31] designed the GSH-bound magnetic nanoparticles which were prepared through the covalent bond linkage of GSH and nanoparticles. It had indicated that this GSH-bound magnetic nanoparticles could retain approximately 87% enzyme activity and obtain glucagon-like peptide-1, a peptide hormone for type 2 diabetes therapy. This design of nano-drug was combining SH in GSH with silla-NH_2_ by covalent bond, as shown in Figure 2B. Mottaghipisheh et al. [32] discovered that *S. marianum*, *B. vulgaris,* and *D. sophia* extracts combining CuO nanoparticles exhibited a certain effect on diabetic rats, and they could significantly decrease the content of GSH-Px to prevent GSH oxidation. Gurunathan research group [33] employed Au nanoparticles (AuNPs) to treat the diabetes and compensated for the loopholes in the body’s antioxidant defense system. The experimental results indicated that the levels of GSH, superoxide dismutase (SOD), catalase and GSH-Px were significantly increased in diabetic rats treated with AuNPs, by inhibition of lipid peroxidation and ROS generation during hyperglycemia. Most of these nano-drugs are active nanoenzymes which directly act on GSH or GSH-Px to regulate GSH synthesis, as shown in Figure 2C. 

### 4.2. Nano-Drug Design Based on the Role of GSH in Polyol Pathway

When the blood glucose concentration in diabetes increases and exceeds the normal metabolic capacity, much glucose is metabolized through the polyol pathway. Aldose reductase (AR) in the polyol pathway reduces the excessive glucose to sorbitol by NADPH as cofactor. A large amount of sorbitol accumulation results in excessive sorbitol in the cell and damages the cell permeability owing to their low lipophilicity. Subsequently, sorbitol does not penetrate the cell membrane, and further cause cell swelling and rupture, inducing a series of diabetes and chronic complications development [34,35,36,37]. GSSH can deplete NADPH and be reduced to GSH by GSH-R. If the synthesis of GSH is normal, or a dramatic decline in the GSH occurs, the NADPH consumption is bound to increase [21]. Hence, the polyol pathway is reversed to restrain the sorbitol production, which provides a new target for the prevention and alleviation of diabetes (Figure 3A).

By competing NADPH with GSH-R and hence resulting in reduced amount of GSH, the polyol pathway increases susceptibility to intracellular oxidative stress. Wang et al. [29] prepared a novel antioxidant GSH liposomes (GSH-LIP) which were applied in therapy of diabetic nephropathy. It indicated that GSH-LIP effectively depleted NADPH to block the polyol pathway, and dramatically relieved diabetic nephropathy, which provided a new theoretical basis for the nano-drug research in therapy diabetic nephropathy. 

## 5. Nano-Drug Design for Tumor Based on the Physiological Properties of GSH

### 5.1. Nanoparticles Implement Tumor Targeting Delivery Mechanisms

#### 5.1.1. Passive Targeting

Passive targeting mainly depends on its nanometer size and the microvascular structure at the tumor site. Compared with normal tissues, most tumor tissues have incomplete vascular remodeling due to vigorous growth and metabolism, with a gap of 10–1000 nm between vascular endothelium. Therefore, nanoparticles of the corresponding size can reach tumor tissues through blood circulation and are enriched in tumor tissues through enhanced permeability and retention (EPR) effect [38]. It is generally believed that 10–100 nm nanoparticles have better EPR effect [39]. On the other hand, the growth state and density of vascular endothelial in tumor area can also affect the EPR effect [40]. 

#### 5.1.2. Active Targeting

In order to further enhance the uptake of nano-drug delivery system by tumor cells, the surface of the nanoparticles can be modified with actively targeted ligand, so that they can enter cells through receptor-ligand-mediated endocytosis by recognizing specific receptors on the surface of tumor cells [41]. Compared with the passive targeting, the active targeting nanoparticles have stronger specificity and can significantly increase the intracellular drug concentration in tumor cells [42].

#### 5.1.3. Tumor Microenvironment Responsive Nano-Drug Delivery System

Compared with normal tissues, tumor tissues and cells present unique characteristics of microenvironment, mainly reflecting the following aspects [43]: (1) pH value: the tumor environment is weakly acidic, pH 6.5–7.0. Tumor cell inclusions or lysosomes have a lower pH of 4.0–6.0 [44]; (2) tumor cells present a reductive environment in which glutathione concentration can reach 1–10 mM, which is 100–1000 times than that of the blood environment [45]; (3) mitochondria of tumor cells present an oxidative environment, in which the concentration of reactive oxygen species (ROS) can reach mM level [46]. pH responsive nano-drug delivery system: the change of the body properties under pH stimulation makes the nanoparticles depolymerize to achieve the purpose of targeted drug delivery in tumor cells [47]. Reductive nano-drug delivery system: according to the concentration difference between GSH in tumor cells and normal tissues, reduction sensitive nanocarrier materials are designed. Disulfide or disselenium bonds contained in the carrier material can be reduced by intracellular GSH and broken, thus causing drastic changes in the properties of the carrier and releasing the encapsulated drugs [48].

### 5.2. Nano-Drug Design Based on NADPH Depletion during GSSG Reduction in Ferroptosis 

Ferroptosis is a programmed cell death pathway that is featured with altered iron and redox homeostasis. Particularity of ferroptosis is generally believed to be the accumulation of ROS relied on iron, resulting in the occurrence of lipid peroxidation and cell death [49]. Furthermore, ferroptosis also shows the decline of regulation of core enzyme GPX4 in antioxidative system (glutathione system). The lipid peroxides will be scavenged by GPX4. If the activity of GPX4 is inhibited, more lipid peroxides will be produced to result in oxidative imbalance and ferroptosis occurrence [50]. Therefore, GPX4 inhibition or modulation of GSH biosynthesis to decrease GPX4 activity are two typical approaches for ferroptosis induction. GSSG is reduced to GSH with GSH-R and consuming NADPH. NADPH is an essential intracellular reducing agent for the elimination of lipid hydroperoxides, and when these processes are impaired, ferroptosis is triggered [51]. In addition, another mechanism of ferroptosis is arachidonic acid/adrenic acid (AA/AdA), in which accumulation of PE-AA-OOH is another apparent marker of ferroptosis. It is worth noting that the accumulation of PE-AA-OOH in cells depends on the activity of GPX4, and PE-AA-OOH can be oxidized to PE-AA-OH in the presence of GPX4 [52,53,54]. Therefore, NADPH depletion, excessive PE-AA-OOH, and GPX4 deficiency are generally proposed as the main characteristics of induced ferroptosis [55,56,57], as shown in Figure 3B. 

**Figure 3 molecules-26-05567-f003:**
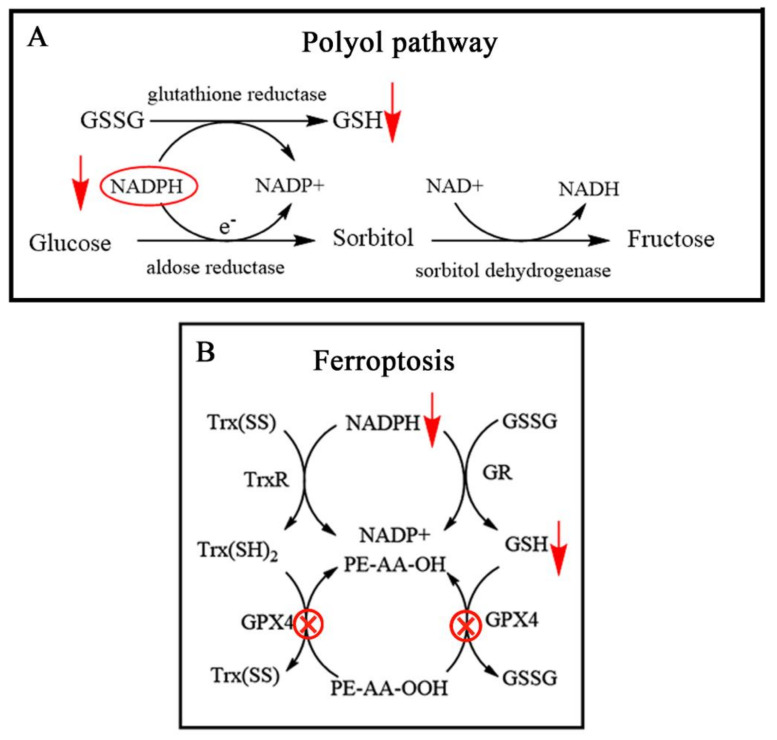
Pathogenesis of GSH involved in: (**A**) mechanism of polyol pathway [21]; (**B**) mechanism of ferroptosis [58].

Wang et al. [58] designed the azobenzene linker with nitroimidazole-conjugated polypeptide (DHM@RSL3), which cleaved under anaerobic environment. DHM@RSL3 nano-micelles entered the cells and cleaved to release RSL3, a kind of GPX4 inhibitor. Meanwhile, azobenzene depletes NADPH, a key coenzyme in the synthesis of GSH and Trx(SH)_2_, resulting in decreasing contents of GSH and Trx(SH)_2_, and dually inducing ferroptosis to promote tumor cell apoptosis. Zhao et al. [59] prepared an RSL3 iron fluorescence inducer, which was encapsulated in micelles to target GPX4. They found that in drug-resistant human ovarian adenocarcinoma cell models, RSL3 micelles were found to be 30 times toxic than the activable control micelles. This is mainly due to a decline in GSH, which enhances the ability of RSL3 to induce ferroptosis. 

### 5.3. Nano-Drug Design Based on GSH Reductive Ability in Tumor Microenvironment

#### 5.3.1. Theory of Redox-Sensitive in Nano-Drug Delivery System

GSH is considered the primary mercaptan-disulfide redox buffer as a reducing agent in cells [60,61]. The concentration of GSH in blood is only 0.1% to 1% of that in cells [62], hence blood is usually the environment in which GSH less mediated redox reactions. However, tumor cells are characterized as abnormal tumor metabolism and elevated GSH level when the oxidative stress produced, and the cytosolic GSH concentration in tumor cells (2–20 mmol·L^−1^) is 1000 times higher than that in normal cells, resulting in presenting strong reducing environment [63,64]. This extreme difference in concentration makes GSH to be a redox trigger in drug delivery system. Therefore, a redox-sensitive targeted nano-drug delivery system has emerged, whose main design feature is the introduction of responsive chemical bonds in the carrier backbone, side chain or crosslinking agent. Moreover, these chemical bonds are relatively stable in the normal environment of the human body including blood and tissue, but they are easy to undergo redox reaction with high concentration of GSH, leading to the cleavage of chemical bonds to release drugs, and achieving accurate delivery of drugs in tumor cells [65,66].

#### 5.3.2. Chemical Bonds That React with GSH

The redox-sensitive chemical bonds play a crucial role in the redox-sensitive targeted nano-drug delivery system, which is equivalent to the switch of the delivery system and directly affects the drug release. There are some common redox-sensitive chemical bonds, such as disulfide bond (-S-S-), mono thioether bond (-S-), conjugate bond of -Pt-O-, diselenide-conjugated bond (-Se-Se-), conjugate bond of -Se-N-, mono selenium bond (-Se-). Among them, disulfide bond has been widely used to develop reduction-responsive drug delivery system for cancer therapy. The kinds and characteristics of common redox-sensitive chemical bonds are shown in Table 1.

#### 5.3.3. Nano-Drug Design Based on Different Chemical Bonds

##### Nano-Drug with S-S 

Disulfide bond (S-S) is one of the most common GSH reduction sensitivity bond, and the main method of introducing -S-S- is to design prodrugs with redox sensitive bonds. Shao et al. [67] successfully combined camptothecin and chlorambucilby by disulfide bonds to design a new drug–drug conjugated prodrug. Under high concentration of GSH in tumor cells, the disulfide bonds are destroyed and effectively release these two anticancer drugs. Compared with a single anticancer drug, two anticancer drugs can not only effectively kill tumor cells, but also notably reduce the adverse side effects on normal cells (Figure 4A). Khorsand et al. [68] designed the thiol-responsive degradable micelles consisting of a pendant disulfide-labeled methacrylate polymer block (PHMssEt) and a hydrophilic poly (ethylene oxide) (PEO) block. The disulfide bond in PEO-b-PHMssEt is cleaved under the action of GSH, leading to the instability of the self-assembled micelles. This GSH-triggered micelle instability altered their size distribution and formed large aggregates, thereby enhancing the release of encapsulated anticancer drugs and providing multifunctional drug delivery applications (Figure 4B). Sun et al. [69] prepared PTX-SS-CIT nanoparticles with higher dual redox sensitivity, faster tumor-specific drug release, and stronger anti-tumor activity (Figure 4C). Luo et al. [70] designed the novel redox responsive conjugates by bridging PTX and OA with disulfide bond (PTX-S-S-OA). PTX-S-S-OA nanoparticles exhibited distinct superiority over both taxol and PTX-OA, and the tumor almost completely disappeared in mice after the treatment by nanoparticles (Figure 4D). Furthermore, there are many nano-drug designs for anti-tumor therapy based on disulfide bond [71,72], which provide a promising perspective for the design of nano-drug delivery system.

##### Nano-Drug with -S- 

Mono thioether bond (-S-) as a binder is widely applied in anti-tumor and nano-drug delivery system design. Cong et al. [73] successfully developed a novel dual redox responsive prodrug-nanosystem (PTX-S-OA/TPGS NPs) assembled by hydrophobic small-molecule prodrugs. PTX-S-OA/TPGS NPs were significantly superior to disulfide conjugate (PTX-2S-OA) in terms of dual redox sensitive drug release and in vivo antitumor efficacy. PTX-S-OA/TPGS NPs have an impressive high drug loading and are effective in selectively releasing drugs at the tumor site, as shown in Figure 5A. Meng et al. [74] synthesized a new prodrug DTX-S-LA, which utilized mono thioether bond as a linker to bridge linoleic acid (LA) and docetaxel (DTX). DTX-S-LA self-assembled with DEPEG-PEG to form nanoparticles with a drug loading capacity of 53.4%. These nanoparticles had the characteristics of uniform particle size, high blood stability, and fast drug release in tumor cells and had higher tumor inhibition rate in vivo compared with free DTX, as shown in Figure 5B. Zhang et al. [75] synthesized a kind of CUR-S-CUR prodrug by coupling two CUR molecules with mono thioether bond for GSH responsive drug delivery, as shown in Figure 5C. These CUR-S-CUR NPs exhibited good colloid stability, more efficient cellular uptake, and intracellular/nuclear drug delivery compared to free CUR.

##### Nano-Drug with Pt-O

Pt-O bond can be reduced and cleaved by GSH to release active metabolite Pt(II). Based on this theory, Ling et al. [76] designed the GSH-sensitive prodrug nanoparticles Pt(IV) for effective drug delivery and cancer therapy. Pt(IV) nano-drugs could resist thiol-mediated detoxification through the GSH depletion. After Pt(IV) nanoparticles are reduced by GSH, Pt-O broke down and released enough active Pt(II) metabolites, which covalently bonded with the target DNA and induced apoptosis (Figure 6A). Huang et al. [77] found that Pt(IV)NP-cRGD exhibited strong echogenic signals and excellent echo persistence under ultrasound imaging. Furthermore, the GSH-sensitive drug delivery system not only maximized the therapeutic effect, but also reduced the toxicity of chemotherapy. Pt(IV)NP-cRGD, together with ultrasound imaging, depleted GSH, and increased ROS levels, leading to mitochondria-mediated apoptosis (Figure 6B).

##### Nano-Drug with Se-Se

Diselenide-conjugated bond (Se-Se) has a unique dual redox sensitivity. High expression of GSH in tumors or ROS generation by oxidative stress, such as H_2_O_2_, can break the diselenide-conjugated bond to complete the redox response. Manjare et al. [78] synthesized a new GSH reduction-triggered fluorescent probe (A) by connecting two molecules of BODIPY-Se by the diselenide-conjugated bond, which could be utilized to detect GSH or H_2_O_2_ in cancer cells. Diselenide-conjugated bond of fluorescent probe (A) was cleaved by GSH, then reacted with ROS to emit fluorescence. Han et al. [79] prepared the fluorescent molecule diselenide SeDSA nanoparticles containing 9, 10-distyrylanthracene (DSA) derivative (SeDSA) with aggregation-induced emission (AIE). SeDSA could co-assemble with the antitumor prodrug and diselenide-containing paclitaxel (SePTX), to form SeDSA-SePTX Co-NPs (Co-NPs). SeDSA-SePTX Co-NPs rapidly disintegrate and release AIE dye and PTX under the reducing environment, which played the role of tumor imaging and tumor therapy. Zhao et al. [80] designed diselenide-crosslinked polymer gels (SeSey-PAA-TPEx) via free radical copolymerization. The diselenide crosslinker in the gels could be fragmented in the presence of H_2_O_2_ or GSH due to its redox-responsive property for diagnosis of tumor. 

##### Nano-Drug with Se-N

Conjugate bond of Se-N is a novel dual redox sensitive bond, which is not only responsive with GSH to form Se-H, but also responsive with H_2_O_2_ to form Se-N, achieving dual redox responsive effect. Xu et al. [81] developed a new dual redox sensitive fluorescent probe (Cy-O-Eb) based on this theory, which could dynamically track the changes of H_2_O_2_ and GSH in living cells, and directly monitored the redox status of cells. The apoptosis process of HepG2 tumor was successfully observed by Cy-O-Eb. In this report, the breakage and generation of Se-N bond in the structure cause a change in fluorescence in the fluorescent probe under two different environments. Under the action of GSH, Se-N bond breaks and generates Se-H structure, and the fluorescence intensity is greatly reduced. On the contrary, the Se-N bond was regenerated and the fluorescence was restored under the effect of H_2_O_2_, as shown in Figure 7. 

##### Nano-Drug with -Se-

Mono selenium bond (-Se-) is an oxidation stimuli responsive bond, which is mainly oxidized by ROS, such as H_2_O_2_, and is ruptured to release nano-drugs. Wang et al. [82] prepared the drug-loaded polymeric nanoparticles of selenium-inserted copolymer (I/D-Se-NPs). I/D-Se-NPs rapidly dissociate in a few minutes mediated by ROS and promoted the continuous release of anti-tumor drugs. Moreover, Jiang et al. [83] developed a dual-stimuli responsive and wormlike micelle system (C_11_-Se-C_11_) using a switchable selenium-containing surfactant. Zhang et al. [84] designed a viscoelastic wormlike micellar solution based on a new redox-responsive surfactant, namely sodium dodecylselanylpropyl sulfate (SDSePS). The above selenium bond in nanoparticles can be oxidized by H_2_O_2_ to form Se=O to play relative activity.

#### 5.3.4. Glutathione Responsive Photodynamic Therapy

Phototherapy can be divided into photothermal therapy (PTT) and photodynamic therapy (PDT). PTT is a treatment method for killing tumors by injecting photothermal materials into the body and irradiating them with near-infrared light (750~1400 nm). When tumor tissues/cells are heated to 40–45 °C, cell membranes and nucleic acids are damaged or mitochondrial dysfunction occurs in the process of hyperthermia. Prolonged exposure to high heat eventually leads to the death of tumor tissue/cells. During PTT, tumor tissue/cells have a lower heat tolerance than normal tissue/cells. Therefore, it is possible to selectively kill tumor tissues/cells by using the ability of local tumor heating, while not harming the normal tissues/cells [85].

PDT has emerged as a technique for disease treatment which requires three essential components: photosensitizers (PS), specific wavelengths of light (ultraviolet light, visible and near-infrared light), and oxygen. Light excitation at a specific site triggers a photochemical reaction in PS resulting in the production of reactive oxygen species (ROS), which subsequently yields tissues/cells damage and death. PDT can provide accurate stimulus that triggers ROS production at a defined time and specific site resulting in a significant reduction of off-target effects on healthy tissues [86,87].

The concentration of intracellular ROS directly determines the effect of photodynamic therapy. Hence, a decline in GSH can increase the level of ROS and promote cell apoptosis, which provides the primary theory for photodynamic therapy. Ruan et al. [88] constructed a nanosystem, Cu-tryptone nanoparticles (Cu-Try NPs), that promoted photodynamic therapy through the consumption of GSH. It demonstrated that Cu-Try NPs could deplete GSH to increase intracellular ROS and improve the photodynamic therapy. Chen et al. [89] developed a kind of hydrophobic cysteine-based polydisulfide amide (Cys-PDSA) polymers and utilized them as black phosphorus quantum dots nanocarrier. Paclitaxel (PTX) was loaded into the nanoparticles to achieve a combination of chemotherapy and photothermal therapy for cancer through GSH reduction mediated by disulfide bond. Yang et al. [90] prepared a new type of pH/GSH multi-response chitosan nanoparticles (SA-CS-NAC), and SA-CS-NAC-loaded photosensitizer ICG to form the amphoteric mercapto chitosan nanoparticles (SA-CS-NAC@ICG NPs) by self-assembly. SA-CS-NAC@ICG NPS successfully achieved multi-response to release ICG in a microenvironment with low pH and high GSH in tumor cells. At the same time, in vitro cell experiments confirmed that SA-CS-NAC@ICG NPS had strong cell uptake ability, low biotoxicity, and good tumor inhibition.

## 6. Nano-Drug Design Based on the Role of GSH in Neurological Diseases

GSH takes participated in the neurodegenerative changes of Parkinson’s disease, mainly against the production of intracellular ROS during oxidative stress. The concentration of GSH in the substantia nigra in patients with Parkinson’s disease dramatically decreased, indicating a close relationship between GSH, oxidative stress, and Parkinson’s disease. Based on the theory above, Ma et al. [91] prepared Ag44(SR)30 silver nanoclusters with a ligand of 5-mercapto-2-nitrobenzoic acid, and completed the high precision detection of GSH, which enables more accurate and comprehensive diagnosis and assessment of Parkinson’s disease. It had been reported that autism spectrum disorders (ASD) were also associated with GSH [92,93,94,95]. The research found that both reduced GSH and total GSH levels were lower in the ASD group than that in the control group [96]. In addition, some studies had found that the treatment with GSH could effectively protect renal tubular epithelial cells, reduce the occurrence of acute renal damage or even acute renal failure, and improve the survival rate of patients with cerebral hemorrhage [97]. Although GSH is directly or indirectly involved in the pathogenesis of neurological diseases, nano-drug design based on the role of GSH in oxidative stress has not been reported. This is a weakness and blind area in the nano science research, we can make full use of the advantages of nanotechnology, combining the characteristics of nervous system diseases to develop new targeted nano-drugs.

## 7. Fluorescent Nano-Probe Design Based on Physiological Properties of GSH

The traditional methods for the visual quantitative determination of intracellular ROS and GSH are mostly instrumental analysis. However, the sample pre-treatment process is complicated, the determination is time-consuming, and the GSH and ROS in vivo cannot be monitored in real time. In contrast, fluorescent probe technology has the advantages of high sensitivity, good selectivity, and good real-time performance, which show the outstanding features for monitoring GSH and ROS in vivo and in vitro [98,99,100]. The following is an introduction to the design of fluorescent nano-probes based on the physiological properties of GSH, hoping to provide some references for the clinical application of nano-probes through the summary of this paper. 

Liu et al. [101] synthesized a novel two-photon fluorescence probe MT-1 for the detection of biological mercaptans mainly GSH in mitochondria. 4-dinitrobenzene sulfonyl group (DNBS) in fluorescent probe, which acted as the responsive group of GSH. Fluorescence of the probe would be quenched due to the electron-absorbing action of DNBS. But when the probe reacted with GSH in mitochondria, DNBS was eliminated, and fluorescence of the probe was restored to directly observe biological mercaptan in living cells and tissues, which were used to detect and observe cell status. Chen et al. [102] prepared a fluorescent probe for the detection of GSH in aqueous solution and living cells by introducing dinitrophenyl ether into 2-(2′-hydroxy-3′-ethoxyphenyl) benzothiazole. The fluorescence of the probe was quenched due to the strong electron-absorbing of the nitro group, but when the probe was reduced by GSH, the fluorophore was released to emit a strong fluorescence at 485 nm. Both of the above designs introduce a strong electron-absorbing group into the probe structure, and the fluorescence of probe is quenched or resurrected after GSH regulation. There are also some references for the application of this design [103,104,105,106,107,108,109].

All the above are small molecular fluorescent probes, and their poor tumor targeted ability and solubility have limited their application in vivo. In order to effectively penetrate tumors, especially those tumors with dense stroma, Niko et al. [110] designed a GSH-responsive fluorescent probe in which the amphiphilic fluorescent material NR12D was self-assembled and coated with a polymer DSP containing disulfide bonds. Li et al. [111] prepared micelles by covalently linking the NIR fluorescent dye dimethyl-4H-pyran (DCM) with the anti-tumor drug gemcitabine using a disulfide bond as a bridge to achieve the targeted positioning and therapeutic effect of the nanoprobe. Zhang et al. [112] synthesized a GSH-responsive probe using the fluorescent material amantadine-naphthalimide and the anticancer drug camptothecin to achieve active fluorescence imaging in cancer cells. Lu et al. [113] used hollow mesoporous carbon (HMC) coated with doxorubicin and grafted reduction-sensitive near-infrared dye (HMC SS-CDPEI) to prepare a nanoprobe to monitor the release of doxorubicin. Choi et al. [114] designed and synthesized a GSH-responsive fluorescent carbon nanoprobe. All these probes disintegrate under the action of GSH, and the fluorescence emission can monitor the drug release in real time.

## 8. Nano-Imaging Design Based on Physiological Properties of GSH

Nano-imaging technology is to design GSH-responsive nanoparticles in which nano-imaging materials are encapsulated in the nanoparticles for dual-mode imaging and combination therapy. Li et al. [115] reported that the drug paclitaxel (PTX) and hydroxyethyl starch were bonded by disulfide bonds, and then the fluorophore DiR was encapsulated in the nanoparticle nucleus during self-assembly, during which the DiR fluorescence was quenched. When the nanoparticles were endocytosed by tumor cells, the disulfide bonds were cleaved by excessive GSH, resulting in the simultaneous release of DiR and PTX in the nanoparticles. The fluorescence of DiR recovered and could be applied in photoacoustic imaging. Yang et al. [116] synthesized a GSH-responsive hyaluronic acid (HA) and poly (ε-caprolactone) copolymer nanoparticle encapsulated with DOX and superparamagnetic iron oxide (SPIO). Under the action of high levels of GSH, disulfide bonds of these nanoparticles broke, releasing internal DOX and SPIO. SPIO could be utilized in magnetic resonance imaging, while DOX was used in chemotherapy, allowing the combination of imaging and chemotherapy. Yang et al. [117] reported that amphiphilic dextran derivatives were developed from disulfide-linked dextran-g-poly-(*N*-3-carbobenzyloxy-L-lysine) graft polymer (Dex-g-SS-PZLL) and used as theranostic nanocarriers for chemotherapy and magnetic resonance imaging. Consequently, these reduction-sensitive nanoparticles are promising theranostic nanocarriers for magnetic resonance imaging and chemotherapy.

## 9. Application of Nanoscale GSH in Food Field

Design of sodium alginate and chitosan bilayer-modified GSH nanoliposomes was reported by Wei et al. [118]. The results of storage stability and gastrointestinal stability showed that sodium alginate and chitosan bilayer-modified liposomes not only enhanced the stability of GSH, but significantly reduced the release rate of GSH in the gastrointestinal tract. Therefore, in a complex food-processing system, the use of sodium alginate and chitosan bilayer-modified liposomes could avoid the rapid release of GSH, increase the stability of GSH, and thus promote the absorption of GSH by gastrointestinal cells, and enhance the nutritional value of food. This study provides a reference basis and data support for the application of GSH nanoliposomes modified by sodium alginate and chitosan in food field.

## 10. Summary and Perspectives

GSH tablets and GSH injection are widely used in clinic. GSH is a kind of polypeptide, which does not exist stably during transportation and preservation, which brings some difficulties for clinical preservation, transportation, and application. Therefore, it is very important to develop nano-drugs and technologies based on the pathological characteristics of GSH so that GSH can play much greater role in clinical practice. However, GSH nanoparticles are limited to basic experiments and have not been widely used in clinical practice. In view of the problems faced by nanotechnology in clinical diseases, it is necessary to design intelligent nanoparticles with the help of interdisciplinary integration. Nanoparticles adjust its chemical and biological functions by stimulating responsive structural changes, so as to realize intelligent biomedical applications, which is a new interdisciplinary research direction.

In conclusion, based on the physiological and pathological properties of GSH, different types of nano-drugs can be designed from the GSH synthesis process and the physiological regulation of GSH, which can not only improve the targeted abilities of nano-drugs, but also achieve the treatment of special diseases. These nanotechnologies take full advantage of the strong reductivity of GSH, the high content of GSH in tumor cells, and the NADPH depletion when GSSH is reduced to GSH designing active targeting nano-drugs. This paper reviews the principles and applications of nano-drugs in diabetes, cancer, nervous system diseases, fluorescent probes, imaging, and food, based on the physiological properties of GSH. These studies make full use of the physiological and pathological value of GSH and develop excellent nano-drug design methods, which provide important scientific significance and application value for the research of related diseases that GSH participates in.

## Figures and Tables

**Figure 1 molecules-26-05567-f001:**
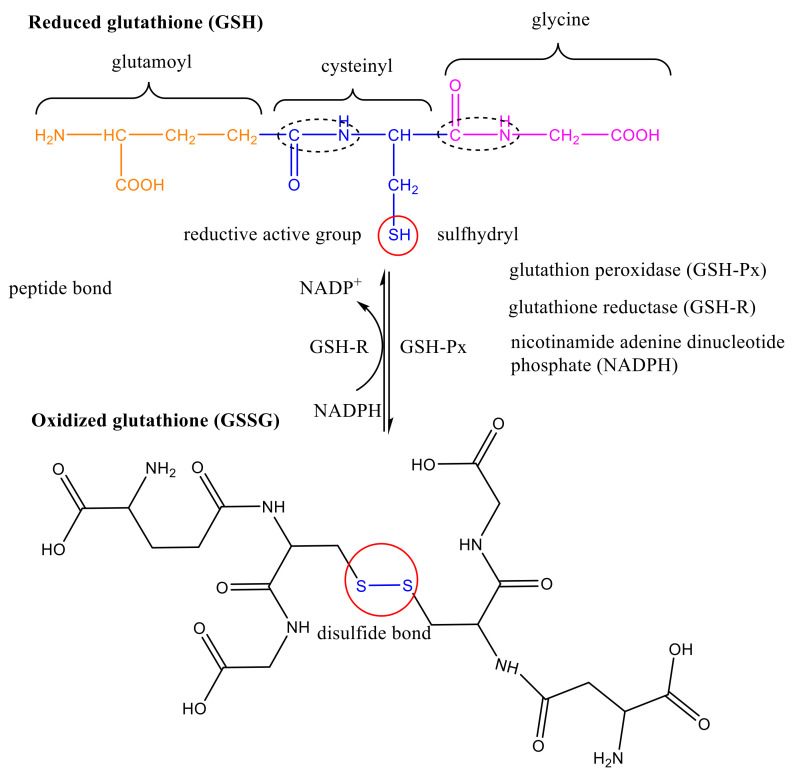
Structure and synthesis procedure of GSH and GSSH.

**Figure 2 molecules-26-05567-f002:**
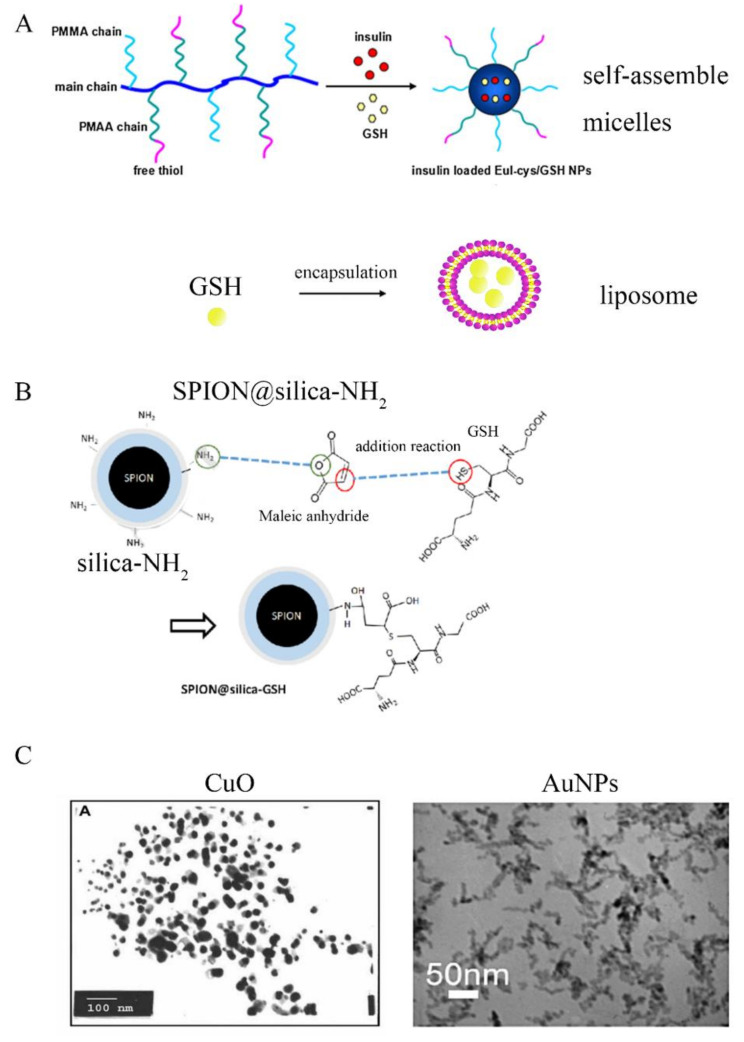
Nano-drugs for diabetes designment based on GSH. (**A**) GSH was encapsulated into Enteric eudragit L100-cysteine to prepare reduced glutathione nanoparticles (Eul-cys/GSH NPs) [30]; (**B**) GSH-bound magnetic nanoparticles (SPION@silica-NH_2_). GSH was reacted with maleic anhydride to form SPION@silica-GSH nanoparticles [31]; (**C**) transmission electron microscope images of CuO nanoparticles and Au nanoparticles enzyme [32,33].

**Figure 4 molecules-26-05567-f004:**
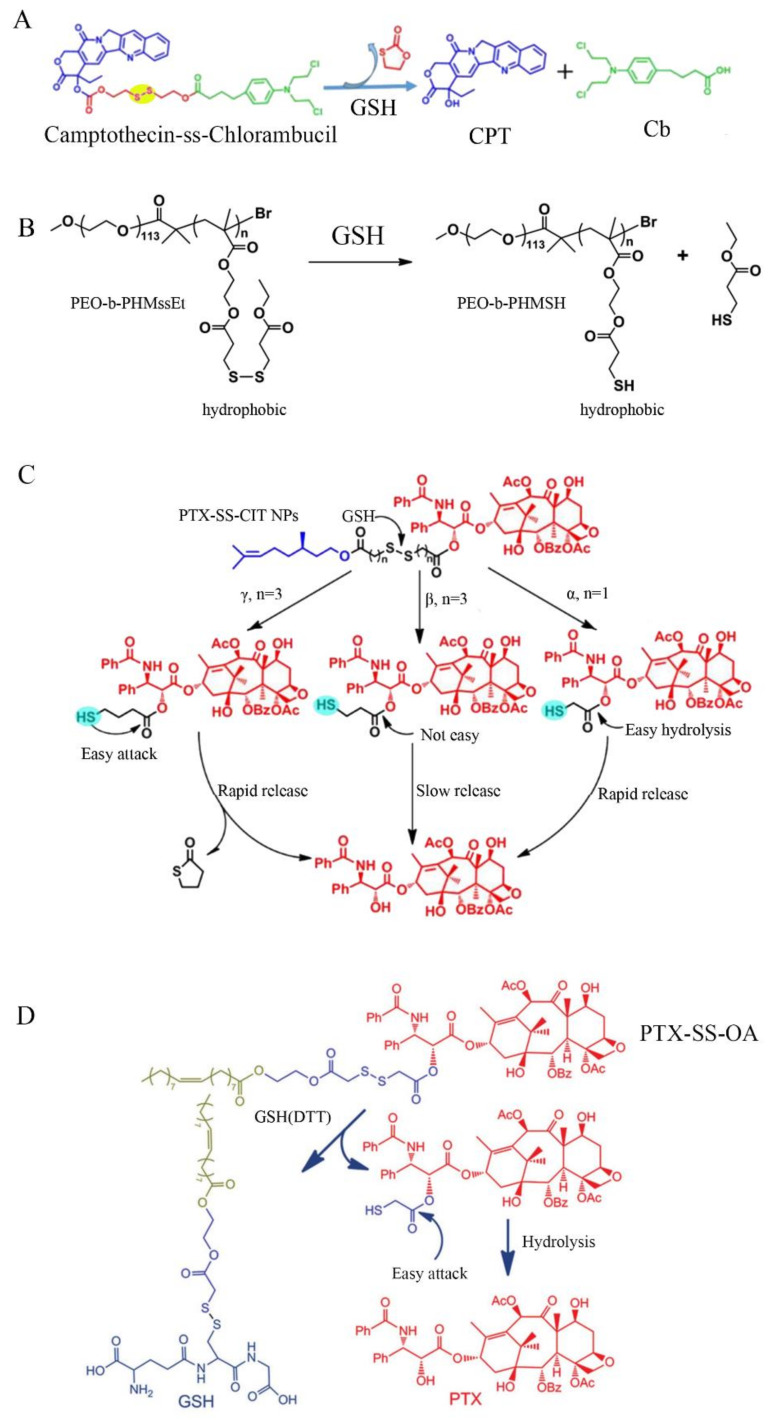
Schematic design of different GSH responsive anticancer drugs with disulfide bond. (**A**) Camptothecin and chlorambucil conjugated with disulfide bond (SS) supramolecular anticancer drugs. Nanoparticles cleavage to CPT with GSH [67]; (**B**) GSH-responsive degradable PEO-b-PHMssEt micelles. PEO-b-PHMssEt cleavage to PEO-b-PHMSH with GSH [68]; (**C**) the disulfide bond-bridged prodrugs PTX-SS-CIT cleavage to different compounds with GSH [60]; (**D**) redox-responsive conjugates by bridging PTX and OA with disulfide bond (PTX-S-S-OA). PTX-S-S-OA cleavage to PTX with GSH [70].

**Figure 5 molecules-26-05567-f005:**
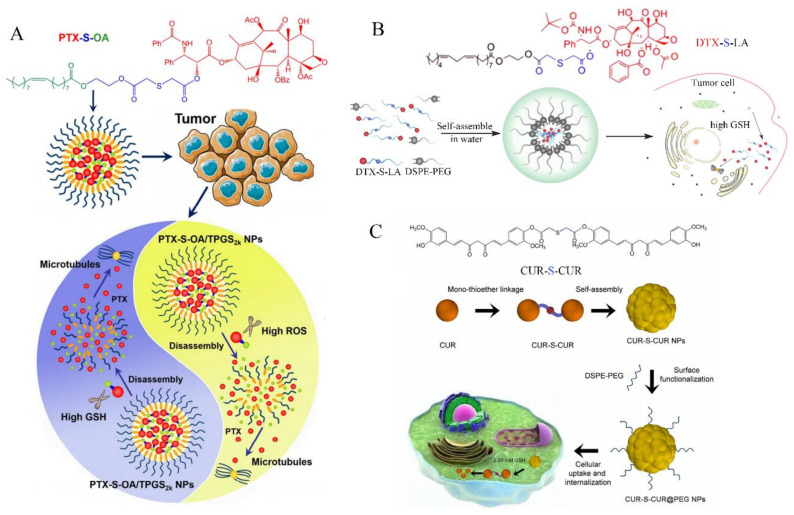
Schematic design of different GSH-responsive anticancer drugs with -S-. (**A**) Schematic representation of the preparation of PEGylated prodrug NPs of PTX-S-OA and cleavage by GSH or ROS [73]; (**B**) schematic representation of DTX-S-LA self-assemble in water and cleavage with GSH in tumor cells [61]; (**C**) schematic representation of CUR-S-CUR prodrug self-assemble and its uptake by tumor cells [75].

**Figure 6 molecules-26-05567-f006:**
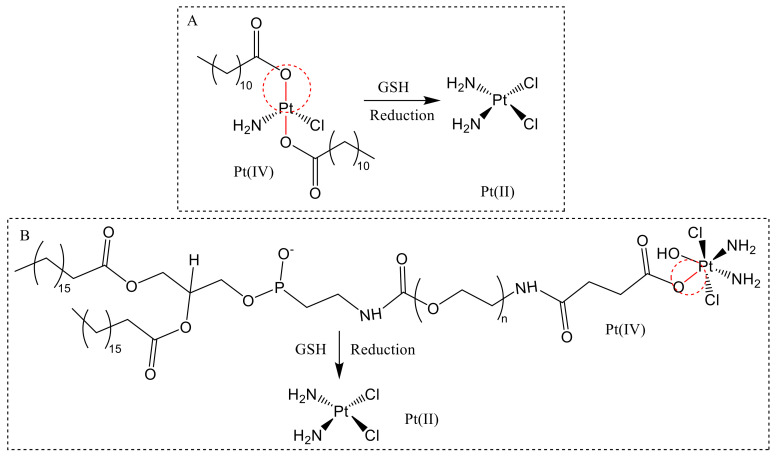
Self-assembled Pt (IV) nanoparticles for specific delivery of Pt drugs. (**A**) Pt (IV) was reduced with GSH to Pt (II) [76]. (**B**) Pt(IV)NP-cRGD was reduced with GSH to Pt (II) [77].

**Figure 7 molecules-26-05567-f007:**
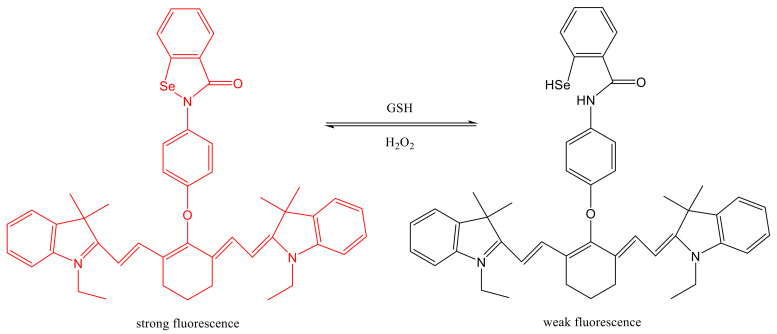
Dual reaction of probe (Cy-O-Eb) with GSH/H_2_O_2_ [81]. The Se-N bond (strong fluorescence) in Cy-O-Eb was reduced with GSH to form Se-H bond (weak fluorescence). Se-N was regenerated and the fluorescence was restored under the effect of H_2_O_2_.

**Table 1 molecules-26-05567-t001:** Redox-sensitive chemical bonds and their characteristics.

Kinds	Response Sensitivity	Stability in Normal Environment	Complexity in Construction	Reference
S-S	strong reduction sensitivity	stable	easy	[67,68,69,70,71,72]
-S-	strong dual redox sensitivity	relatively stable	relatively easy	[73,74,75]
Pt-O	strong reduction sensitivity	relatively stable	easy	[76,77]
Se-Se	strong dual redox sensitivity	relatively stable	easy	[78,79,80]
Se-N	strong dual redox sensitivity	relatively stable	easy	[81]
-Se-	weak dual redox sensitivity	unstable	relatively easy	[82,83,84]

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
