# Peer review of "Nano-Drug Design Based on the Physiological Properties of Glutathione"

_molecules, 2021, doi:10.3390/molecules26185567_

Round 1

Reviewer 1 Report

Dear authors,

In general, the review is properly focused on glutathione and it is mentioned the potential use of glutathione as microenvironment-responsive molecule able to induce the release of other compounds included in the nanoforumations. However, this review is not structured properly and I believe the authors should include a small chapter (between chapter 2 and 3) to discuss nanoparticles in general as delivery systems since there is no proper introduction to the field of nanotechnology. At the beginning of chapter 4, I believe the authors should include a general description on how nanoparticles can reach the tumor (EPR effect for example) and explain some of the characteristic of the tumor microenvironment used to induce the release of cargo molecules from nanoparticles (for example the acidic or hypoxic microenvironment). This should help the reader to understand the nanoformulations described in the following paragraphs. For example, paragraphs 4.2.1 and 4.2.2 should be introduced earlier since these are key topics of this manuscript. The authors introduced many concepts without giving explanation or a description. For example, in chapter 4.2.4 they mention photodynamic and photothermal therapies without explaining what they are. In the last chapter, the author should mention some problematics related to the delivery of nanotherapeutics and their clinical translation. Lastly, all the figures proposed are reproductions from other manuscripts and are very difficult to understand since no proper description is provided in the figure legends.

The review is not easy to read and I’ve noticed many English mistakes (some of them highlighted below). I believe the manuscript should be checked and edited by a native English speaker.

Page 1, Line 21: (GSH) was discovered

Page 2, Line 51: including its inability

Page 2, Line 55: other conjugation methodologies

Page 2, Line 57-59: this phrase need rewording since it is not understandable

Page 4, Line 115: sorbitol does not penetrate the cell membrane effectively

References:

In general, some of the references used are not helpful in consolidating the phrase they are referred to. I believe all the references should be double checked again to make sure they are relevant to the text. In addition, some of the references used are quite old and should be replaced by more recent literature. I have highlighted some problems in the references below.

11: this review is focused on yeast and microorganisms; thus, I don’t believe is suited in the context on this phrase

13: this review is focused on monocytes and not neuronal excitatory intoxication

14-16: these papers are focused on a neurotoxicity while they are mentioned in the context of cancer and chemotherapy

17-19: these references seem very specific while you are describing the effect of glutathione in general. I would reference only 1 recent and broad review on glutathione

30-33: these manuscripts do not describe nanoformulations?

50-52: these manuscripts are related to nucleic acid delivery which are not mentioned in the text?

Author Response

Dear Editor:

Thank you and the reviewers for your valuable suggestions. We have carefully read through the comments and made proper revisions. In addition, the writing was edited by an English editor. Our responses to the reviewer’s questions are listed below. We greatly appreciate your time and efforts to improve our manuscript for publication.

Sincerely,

Wenhua Li

In general, the review is properly focused on glutathione and it is mentioned the potential use of glutathione as microenvironment-responsive molecule able to induce the release of other compounds included in the nanoforumations.

 However, this review is not structured properly and I believe the authors should include a small chapter (between chapter 2 and 3) to discuss nanoparticles in general as delivery systems since there is no proper introduction to the field of nanotechnology.

Response: a small chapter (between chapter 2 and 3) to discuss nanoparticles in general as delivery systems have been added in the manuscript in blue. (line 63-80).

  1. Nano-drug delivery systems

With the high investment and rapid development in recent years, nanotechnology has applied in all fields of biomedical science and technology [17]. Similarly, nanotechnology provides new approach for drug delivery, especially targeted drug delivery. Targeted drug delivery systems deliver desired drugs to the diseased parts and reduce distribution to normal tissues or cells [18]. Advantages of nanoparticles as drug delivery are described follow: (1) dissolve insoluble drugs and prevent drug degradation from the body; (2) prolong the circulation time of drugs; (3) exhibit good biocompatibility and biodegradability; (4) possess high drug loading capacity and low toxicity; (4) selectively deliver drugs to therapeutic targets, such as tumor tissue, tumor cells, tumor-associated stromal cells, and suborganelles [19]. So far, numerous materials such as polymers, lipids and inorganic materials have been developed and used as drug carriers to control the release behavior of drugs [20-21]. In addition, REDOX response stimulation has been highly valued in the treatment of disease and is widely used in nanomedical drug delivery [22, 23]. REDOX potentials in microenvironments are multivariable in different tissues and can be used to design REDOX sensitive delivery systems. Therefore, the design and manufacture of glutathione responsive nanoparticles may be a promising approach for targeted drug delivery [24].

At the beginning of chapter 4, I believe the authors should include a general description on how nanoparticles can reach the tumor (EPR effect for example) and explain some of the characteristic of the tumor microenvironment used to induce the release of cargo molecules from nanoparticles (for example the acidic or hypoxic microenvironment).

Response: these relative contents have been added in the manuscript in blue. (line 148-180).

  1. Nano-drug design for tumor based on the physiological properties of GSH

5.1 Nanoparticles implement tumor targeting delivery mechanisms

5.1.1 Passive targeting

Passive targeting mainly depends on its nanometer size and the microvascular structure at the tumor site. Compared with normal tissues, most tumor tissues have incomplete vascular remodeling due to vigorous growth and metabolism, with a gap of 10-1000 nm between vascular endothelium. Therefore, nanoparticles of the corresponding size can reach tumor tissues through blood circulation and are enriched in tumor tissues through enhanced permeability and retention (EPR) effect [38]. It is generally believed that 10-100 nm nanoparticles have better EPR effect [39]. On the other hand, the growth state and density of vascular endothelial in tumor area can also affect the EPR effect [40].

5.1.2 Active targeting

In order to further enhance the uptake of nano-drug delivery system by tumor cells, the surface of the nanoparticles can be modified with actively targeted ligand, so that they can enter cells through receptor-ligand-mediated endocytosis by recognizing specific receptors on the surface of tumor cells [41]. Compared with the passive targeting, the active targeting nanoparticles have stronger specificity and can significantly increase the intracellular drug concentration in tumor cells [42].

5.1.3. Tumor microenvironment responsive nano-drug delivery system

Compared with normal tissues, tumor tissues and cells present unique characteristics of microenvironment, mainly reflect in the following aspects [43]: (1) pH value: the tumor environment is weakly acidic, pH6.5-7.0. Tumor cell inclusions or lysosomes have a lower pH of 4.0-6.0 [44]; (2) Tumor cells present a reductive environment in which glutathione concentration can reach 1-10 mM, which is 100-1000 times than that of the blood environment [45]; (3) Mitochondria of tumor cells present an oxidative environment, in which the concentration of reactive oxygen species (ROS) can reach mM level [46]. pH responsive nano-drug delivery system: the change of the body properties under pH stimulation makes the nanoparticles depolymerize to achieve the purpose of targeted drug delivery in tumor cells [47]. Reductive nano-drug delivery system: according to the concentration difference between GSH in tumor cells and normal tissues, reduction sensitive nanocarrier materials are designed. Disulfide or disselenium bonds contained in the carrier material can be reduced by intracellular GSH and broken, thus causing drastic changes in the properties of the carrier and releasing the encapsulated drugs [48].

This should help the reader to understand the nanoformulations described in the following paragraphs. For example, paragraphs 4.2.1 and 4.2.2 should be introduced earlier since these are key topics of this manuscript.

Response: first thanks for your insightful comments. 4.1 introduces the role of GSH in ferroptosis and nano-drugs directly or indirectly reduce GSH to achieve anti-tumor ability. However, 4.2 introduces the nanomaterials which were broken with GSH to achieve the effect of tumor treatment. “4.2.1. Theory of redox-sensitive in nano-drug delivery system.” and “4.2.2. Chemical bonds that react with GSH” both introduce the principle of GSH fracture nanoparticles. Therefore, I think it's better to put these contents after 4.2.

The authors introduced many concepts without giving explanation or a description. For example, in chapter 4.2.4 they mention photodynamic and photothermal therapies without explaining what they are.

Response: the concepts of photodynamic and photothermal therapies have been added in the in chapter 4.2.4. (line 354-368).

Phototherapy can be divided into photothermal therapy (PTT) and photodynamic therapy (PDT). PTT is a treatment method for killing tumors by injecting photothermal materials into the body and irradiating them with near-infrared light (750~1400 nm). When tumor tissues/cells are heated to 40-45°C, cell membranes and nucleic acids will be damaged or mitochondrial dysfunction will occur in the process of hyperthermia. Prolonged exposure to high heat eventually leads to the death of tumor tissue/cells. During PTT, tumor tissue/cells have a lower heat tolerance than normal tissue/cells. Therefore, it is possible to selectively kill tumor tissues/cells by using the ability of local tumor heating, while not causing harm to normal tissues/cells [85].

PDT has emerged as a technique for disease treatment which requires three essential components: photosensitizers (PS), specific wavelengths of light (ultraviolet light, visible and near-infrared light), and oxygen. Light excitation at a specific site triggers a photochemical reaction in PS resulting in the production of reactive oxygen species (ROS), which subsequently yields tissues/cells damage and death. PDT can provide accurate stimulus that triggers ROS production at a defined time and specific site resulting in a significant reduction of off-target effects on healthy tissues [86,87].

In the last chapter, the author should mention some problematics related to the delivery of nanotherapeutics and their clinical translation.

Response: in the last chapter, I have added some problems and contents about the delivery of nanotherapeutics and their clinical translation. (line 476-487)

  1. Summary and perspectives

GSH tablets and GSH injection are widely used in clinic. GSH is a kind of polypeptide, which does not exist stablely during transportation and preservation, which brings some difficulties for clinical preservation, transportation and application. Therefore, it is very important to develop nano-drugs and technologies based on the pathological characteristics of GSH so that GSH can play much greater role in clinical practice. However, GSH nanoparticles are limited to basic experiments and have not been widely used in clinical practice. In view of the problems faced by nanotechnology in clinical diseases, it is necessary to design intelligent nanoparticles with the help of interdisciplinary integration. Nanoparticles adjust its chemical and biological functions by stimulating responsive structural changes, so as to realize intelligent biomedical applications, which is a new interdisciplinary research direction.

Lastly, all the figures proposed are reproductions from other manuscripts and are very difficult to understand since no proper description is provided in the figure legends.

Response: some figure legends have been added to describe again in detail and made the figures much clear for readers.

Figure 2. Nano-drugs for diabetes designment based on GSH. A: GSH was encapsulated into Enteric eudragit L100-cysteine to prepare reduced glutathione nanoparticles (Eul-cys/GSH NPs) [31]; B: GSH-bound magnetic nanoparticles (SPION@silica-NH2). GSH was reacted with maleic anhydride to form SPION@silica-GSH nanoparticles [32]; C: Transmission electron microscope images of CuO nanoparticles and Au nanoparticles enzyme [33,34].

Figure 4. Schematic design of different GSH responsive anticancer drugs with disulfide bond. A: Camptothecin and chlorambucil conjugated with disulfide bond (SS) supramolecular anticancer drugs. Nanoparticles would be cleaved with GSH to CPT [68]; B: GSH-responsive degradable PEO-b-PHMssEt micelles. PEO-b-PHMssEt would be cleaved with GSH to PEO-b-PHMSH [69]; C: The disulfide bond-bridged prodrugs PTX-SS-CIT would be cleaved with GSH to different compounds [61]; D: Redox-responsive conjugates by bridging PTX and OA with disulfide bond (PTX-S-S-OA). PTX-S-S-OA would be cleaved with GSH to PTX [71].

Figure 5. Schematic design of different GSH responsive anticancer drugs with -S-. A: Schematic representation of the preparation of PEGylated prodrug NPs of PTX-S-OA and be cleaved by GSH or ROS [74]; B: Schematic representation of DTX-S-LA self-assemble in water and be cleaved with GSH in tumor cells [62]; C: Schematic representation of CUR-S-CUR prodrug self-assemble and be uptaken by tumor cells [76].

Figure 6. Self-assembled Pt (IV) nanoparticles for specific delivery of Pt drugs. A: Pt (IV) was reduced with GSH to Pt (II)[64]. B: Pt(IV)NP-cRGD was reduced with GSH to Pt (II) [65].

Figure 7. Dual reaction of probe (Cy-O-Eb) with GSH/H2O2 [69]. The Se-N bond (strong fluorescence) in Cy-O-Eb was reduced with GSH to form Se-H bond (weak fluorescence). Se-N was regenerated and the fluorescence was restored under the effect of H2O2.

The review is not easy to read and I’ve noticed many English mistakes (some of them highlighted below). I believe the manuscript should be checked and edited by a native English speaker.

Response: the manuscript has been checked and revised line by line by some native English speaker.

Page 1, Line 21: (GSH) was discovered

Line 21: “is” has been revised to “was”.

Page 2, Line 51: including its inability

Line 51: “including of inability” has been revised to “including its inability”.

Page 2, Line 55: other conjugation methodologies

Line 55: “methods” has been revised to “other conjugation methodologies”

Page 2, Line 57-59: this phrase need rewording since it is not understandable

Line 57-59: this sentence has been revised to “In addition, the strong reduction characteristic of GSH in microenvironment can be used to cleave specific redox responsive nanoparticles in order to achieve effects of controlled-release and target of drugs.”

Page 4, Line 115: sorbitol does not penetrate the cell membrane effectively

Line 115: “sorbitol is difficult to penetrate the cell membrane” has been revised to “sorbitol does not penetrate the cell membrane,”.

References:

In general, some of the references used are not helpful in consolidating the phrase they are referred to. I believe all the references should be double checked again to make sure they are relevant to the text. In addition, some of the references used are quite old and should be replaced by more recent literature. I have highlighted some problems in the references below.

11: this review is focused on yeast and microorganisms; thus, I don’t believe is suited in the context on this phrase

Response: reference [11] has been replaced by some relative references [11-13].

[11] Agarwal P. Assessment of Anti-aging Efficacy of the Master Antioxidant Glutathione. International Journal of Sciences: Basic and Applied Research (IJSBAR) 2017, 33, 257-265.

[12] Anaisa V. Ferreira A.V.; Valerie A. C. M. Koeken V.C.; Matzaraki V.; Kostidis S.; Alarcon-Barrera J.C.; de Bree L. C., Moorlag S.J.; Mourits V.P.; Boris Novakovic B.; Giera M.A.; Netea M.G.; Domínguez-Andrés J. Glutathione Metabolism Contributes to the Induction of Trained Immunity. Cells 2021, 10, 971.

[13] Vanin A.F.; Ostrovskaya L.A., Korman D.B.; Kubrina L.N.; Borodulin R.R.; Fomina M.M.; Bluchterova N.V.; Rykova V.A.; Timoshin A.A. Anti-Tumour Activity of Dinitrosyl Iron Complex with Glutathione and S-Nitrosoglutathione Preparations: Comparative Studies. Biofizika 2015, 60, 1157.

13: this review is focused on monocytes and not neuronal excitatory intoxication

Response: the reference [13] has been replace by [15].

[15] Escartin C.; Joon Won S.J.; Malgorn C.; Auregan G.; Berman A.E.; Chen P.C., De´glon N.; Johnson J.A.; Suh S.W.; Raymond A. Swanson R.A. Nuclear Factor Erythroid 2-Related Factor 2 Facilitates Neuronal Glutathione Synthesis by Upregulating Neuronal Excitatory Amino Acid Transporter 3 Expression. The Journal of Neuroscience 2011, 31, 7392–7401.

14-16: these papers are focused on a neurotoxicity while they are mentioned in the context of cancer and chemotherapy

Response: the references [14-16] has been replaced by reference [16].

[16] Zhang L.W.; Li X.L.; Zhao H.B.; Lie M.J.; Li Z.H. Influence of glutathione responsive tumor-targeted camptothecin nanoparticles on glioma based on oxidative stress. Colloid and Interface Science Communications 2021, 42, 100423.

17-19: these references seem very specific while you are describing the effect of glutathione in general. I would reference only 1 recent and broad review on glutathione

Response: first thanks for your insightful comments. I agree with you, but I think more detailed references can make the article much clearer for readers. Therefore, I retain the reference [17] and [18], while reference [19] (now [25-27]) has been replaced by [27].

[27] Raj Rai S.R.; Bhattacharyya C.; Sarkar A.; Chakraborty S.; Sircar E.; Dutta S. Glutathione: Role in Oxidative/Nitrosative Stress, Antioxidant Defense, and Treatments. ChemistrySelect 2021, 6, 4566-4590.

30-33: these manuscripts do not describe nanoformulations?

Response: references [30-33] have been carefully checked. They don’t describe the nanoparitcles. Therefore, I have deleted them.

50-52: these manuscripts are related to nucleic acid delivery which are not mentioned in the text?

Response: I have carefully checked the reference 50-52. It is not directly related to the content of the article. Therefore, I decided to delete these references.

Reviewer 2 Report

Comments to molecules-1364270

This review entitled “Nano-drug design based on the physiological properties of glutathione” by Li, et al, first introduces the Glutathione from the structure to the physiological function, and then thoroughly reviews the applications in nano-drug design for diabetes, tumor, neurological diseases, fluorescent nano-probe design, nano-imaging design, and food field based on the physiological properties. It is of interest to the Molecules community, especially to those researchers who are interested in utilizing the physiological properties of glutathione to study related diseases. This review is well organized and written. However, if the summary and perspectives include more discussions in terms of, but not limited to, e.g., current issues in the application of GSH based on the physiological properties and potential applications in other areas, this review will provide more scientific significance to the related researchers. Therefore, I will recommend the publication after the authors address this minor issue.

Author Response

Dear Editor:

Thank you and the reviewers for your valuable suggestions. We have carefully read through the comments and made proper revisions. In addition, the writing was edited by an English editor. Our responses to the reviewer’s questions are listed below. We greatly appreciate your time and efforts to improve our manuscript for publication.

Sincerely,

Wenhua Li

Comments and Suggestions for Authors

Comments to molecules-1364270

This review entitled “Nano-drug design based on the physiological properties of glutathione” by Li, et al, first introduces the Glutathione from the structure to the physiological function, and then thoroughly reviews the applications in nano-drug design for diabetes, tumor, neurological diseases, fluorescent nano-probe design, nano-imaging design, and food field based on the physiological properties. It is of interest to the Molecules community, especially to those researchers who are interested in utilizing the physiological properties of glutathione to study related diseases. This review is well organized and written. However, if the summary and perspectives include more discussions in terms of, but not limited to, e.g., current issues in the application of GSH based on the physiological properties and potential applications in other areas, this review will provide more scientific significance to the related researchers. Therefore, I will recommend the publication after the authors address this minor issue.

Response: I have revised the summary and perspectives as the requirement. (line 476-486)

  1. Summary and perspectives

GSH tablets and GSH injection are widely used in clinic. GSH is a kind of polypeptide, which does not exist stablely during transportation and preservation, which brings some difficulties for clinical preservation, transportation and application. Therefore, it is very important to develop nano-drugs and technologies based on the pathological characteristics of GSH so that GSH can play much greater role in clinical practice. However, GSH nanoparticles are limited to basic experiments and have not been widely used in clinical practice. In view of the problems faced by nanotechnology in clinical diseases, it is necessary to design intelligent nanoparticles with the help of interdisciplinary integration. Nanoparticles adjust its chemical and biological functions by stimulating responsive structural changes, so as to realize intelligent biomedical applications, which is a new interdisciplinary research direction.

Round 2

Reviewer 1 Report

Dear Authors,

You have carefully adressed my concerns and I believe your manuscript is now ready for publication.

Good luck with your future research endeavors.

Regards